# Continually recruited naïve T cells contribute to the follicular helper and regulatory T cell pools in germinal centers

Julia Merkenschlager [1] ✉, Riza-Maria Berz[1], Victor Ramos[1], Maximilian Uhlig[1], Andrew J. MacLean[1], Carla R. Nowosad [2], Thiago Y. Oliveira [1] & Michel C. Nussenzweig [1,3]

Follicular helper T cells (T$_{FH}$) mediate B cell selection and clonal expansion in germinal centers (GCs), and follicular regulatory T cells (T$_{FR}$) prevent the emergence of self-reactive B cells and help to extinguish the reaction. Here we show that GC reactions continually recruit T cells from both the naïve conventional and naive thymic regulatory T cell (Treg) repertoires. In the early GC, newly recruited T cells develop into T$_{FH}$, whereas cells entering during the contraction phase develop into T$_{FR}$ cells that contribute to GC dissolution. The T$_{FR}$ fate decision is associated with decreased antigen availability and is modulated by slow antigen delivery or mRNA vaccination. Thus, invasion of ongoing GCs by newly developing T$_{FH}$ and T$_{FR}$ helps remodel the GC based on antigen availability.

Germinal centers (GCs) are transient microanatomic structures that support B cell clonal expansion and affinity maturation[1–7]. Selected B cells enter the GC based on their relative affinity for antigen[5]. Within the GC, they undergo iterative rounds of rapid cell division accompanied by antibody gene mutation and affinity selection. This selection is mediated by limiting numbers of GC T follicular helper (T$_{FH}$) cells that provide trophic signals to B cells based on their ability to capture and present antigen in the form of peptide-major histocompatibility complex (p-MHCII)[8–10]. The relative proportion of T$_{FH}$ help provided is dependent on the amount of cognate peptide-major histocompatibility complex (p-MHC) presented by the B cell[11]. Thus, B cells with the highest affinity antigen receptors that can capture and present the most antigens are selected by T$_{FH}$ cells for clonal expansion[9,11–15]. Beyond clonal expansion, T$_{FH}$ cells also coordinate the differentiation of GC B cells into plasma cells and memory B cells[13,16–18].

In addition to T$_{FH}$ cells, GCs also contain variable numbers of T$_{FR}$ cells. These regulatory cells express Foxp3, accumulate during the contraction phase of the GC, and suppress the development of B cells producing Immunoglobulin E (IgE) and self-reactive antibodies[19–26].

Although the equilibrium between conventional T$_{FH}$ and regulatory T$_{FR}$ plays a crucial role in regulating autoimmunity and controlling the size, longevity, and products of the GC response, how this is modulated is not understood.

Like B cells, T$_{FH}$ cells undergo clonal expansions and selection in the GC[27]. GCs are open structures that recruit new B cells throughout the immune response and also support differentiated T$_{FH}$ cell migration between established GCs[28,29]. Whether GCs also remain open to naïve T cell entry throughout the reaction and how continual recruitment might alter the balance between T$_{FH}$ and T$_{FR}$ remains to be determined[29].

Here we use lineage tracing to fate map naïve T cells and follow their differentiation into T$_{FH}$ and T$_{FR}$ during an ongoing immune response. The data reveals that the GC T cell compartment is continually enriched by newly entering T cells that subsequently undergo clonal expansion. Notably, the invading T cells develop primarily into conventional helper T$_{FH}$ in the early stages of the GC response. However, overtime the relative proportion of invading T cells that develop into regulatory T$_{FR}$ increases in a manner that is directly related to decreasing antigen availability.

[1]Laboratory of Molecular Immunology, The Rockefeller University, New York, NY 10065, USA. [2]Translational Immunology Center, New York University Grossman School of Medicine, New York, NY 10016, USA. [3]Howard Hughes Medical Institute, The Rockefeller University, New York, NY 10065, USA. ✉e-mail: jmerkensch@rockefeller.edu

## Results

To determine whether naïve T cells can continue to differentiate into $T_{FH}$ cells (CD4$^+$CD62L$^{low}$ CXCR5$^{high}$PD1$^{high}$) during the GC response, we utilized *Sell*CreERT2 ROSAtdTomato mice (CD62L reporter mice). In these mice, tamoxifen-inducible Cre is under the control of regulatory elements of *CD62L*, a gene that is expressed in naïve and recently activated T cells but not in differentiated $T_{FH}$ cells (Supplementary Fig. 1A–D)[27]. Up to 75% of the naïve T cell compartment but less than 1% of the $T_{FH}$ in the spleens, mesenteric lymph nodes, and Peyer's patches of un-immunized CD62L reporter mice were permanently labeled 3 days after tamoxifen exposure (Supplementary Fig. 1B–D). To further document the specificity of the CD62L reporter mice, we followed the kinetics of labeling after immunization. Tamoxifen was administered on day 10 after Ovalbumin (OVA) immunization, and tdTomato expression was monitored in the naive and $T_{FH}$ compartments thereafter (Supplementary Fig. 1E–G). Indicator expression was initially detected in the naïve T cell compartment after 12 h, but the label was not appreciably detected in the $T_{FH}$ compartment until 3 days later (Supplementary Fig. 1E–G). Thus, the label accumulates in the $T_{FH}$ compartment later but in parallel to the naïve compartment suggesting a precursor product relationship.

$T_{FH}$ clones disseminate equally throughout GCs in the spleen as documented in experiments in which half spleens were assayed independently[27]. Therefore to document the dynamics of $T_{FH}$ responses over time, we examined the repertoire of $T_{FH}$ cells in

immunized animals longitudinally by performing sequential splenic biopsies[27]. To focus on immunization-induced T cells and to eliminate preexisting $T_{FH}$ repertoires we labeled the naïve compartment by administering tamoxifen 4 days before immunization with 4-hydroxy-3-nitrophenylacetyl-conjugated ovalbumin (NP-OVA) (Supplementary Fig. 2A). Spleen biopsy was performed on days 7 and 21 after immunization and single cell, paired T cell receptor (TCR) alpha and -beta chain sequences were obtained from purified tdTomato labeled $T_{FH}$ cells (Fig. 1A–C, Supplementary Fig. 2B–D). Here, naïve cells initiating responses from the point of immunization will be marked by tdTomato label, allowing them to be distinguished from pre-existing $T_{FH}$ that occur spontaneously in the spleen[30]. TCR-repertoire sequencing showed that extensive clonotypic evolution occurred between days 7 and 21 of immunization, with over half of the clones present on day 21 expressing receptors not detected on day 7 (Fig. 1A–C, Supplementary Fig. 2B–D)[27]. We reasoned that the novel clones found on day 21 could either be expanded T cell clones not detected on day 7 or alternatively be due to late entry of newly activated T cells arriving into the $T_{FH}$ compartment.

To determine whether T cells continue to enter the $T_{FH}$ compartment during an immune response, the CD62L expressing compartment in reporter mice was labeled on day 7 after immunization by tamoxifen delivery. $T_{FH}$ populations obtained from draining popliteal lymph nodes were then assayed on day 21(Fig. 1D, E, Supplementary Fig. 3A, B). The rationale was that if labeled naïve T cells subsequently

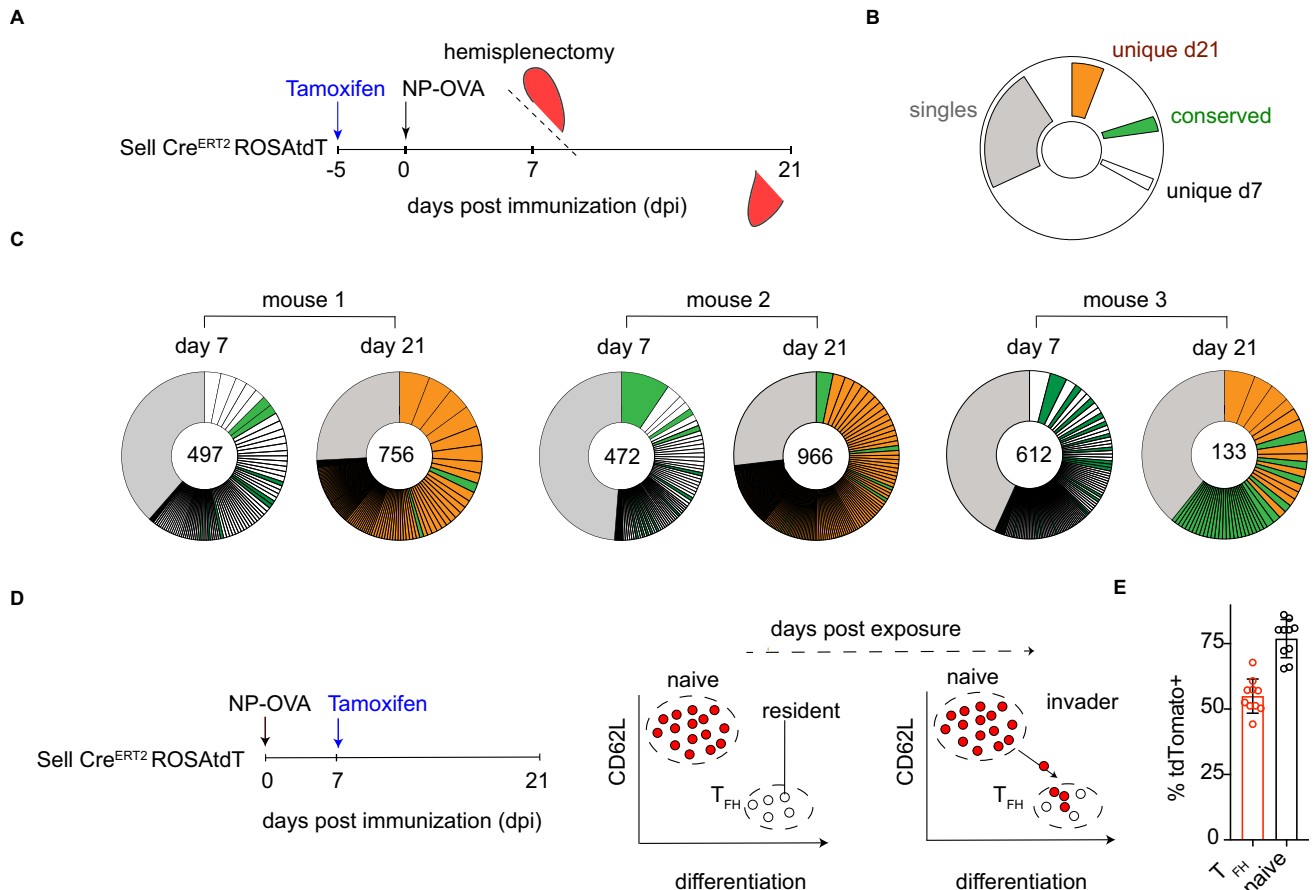

**Fig. 1 | Extensive clonal evolution of the $T_{FH}$ response over time. A** Schematic representation of the experimental setup in (**C**). **B** Color-coded indexing for the clonal behaviors between day (d) 7 and d21 post immunization in (**C**). Conserved TCR clonotypes are represented in green, those found only on d7 in white, novel clones appearing only on d21 in orange, and singles in gray. **C** Pie charts show clones of $T_{FH}$ cells in each mouse at the indicated time. Segments are proportional to the

representation of each clone. Numbers inside the pie charts indicate the number of TCR sequences illustrated. GEO Submission (GSE147182). **D** Schematic (left) and diagrammatic (right) representations of the experimental setup and the fate labeling systems used in (**E**). **E** Graph shows the frequency of tdTomato labeled cells in the $T_{FH}$ or in the naïve T cell compartment on days 21 post immunization and 14 after tamoxifen exposure; *n* = 10 mice per group. Data are presented as mean value ± SD.

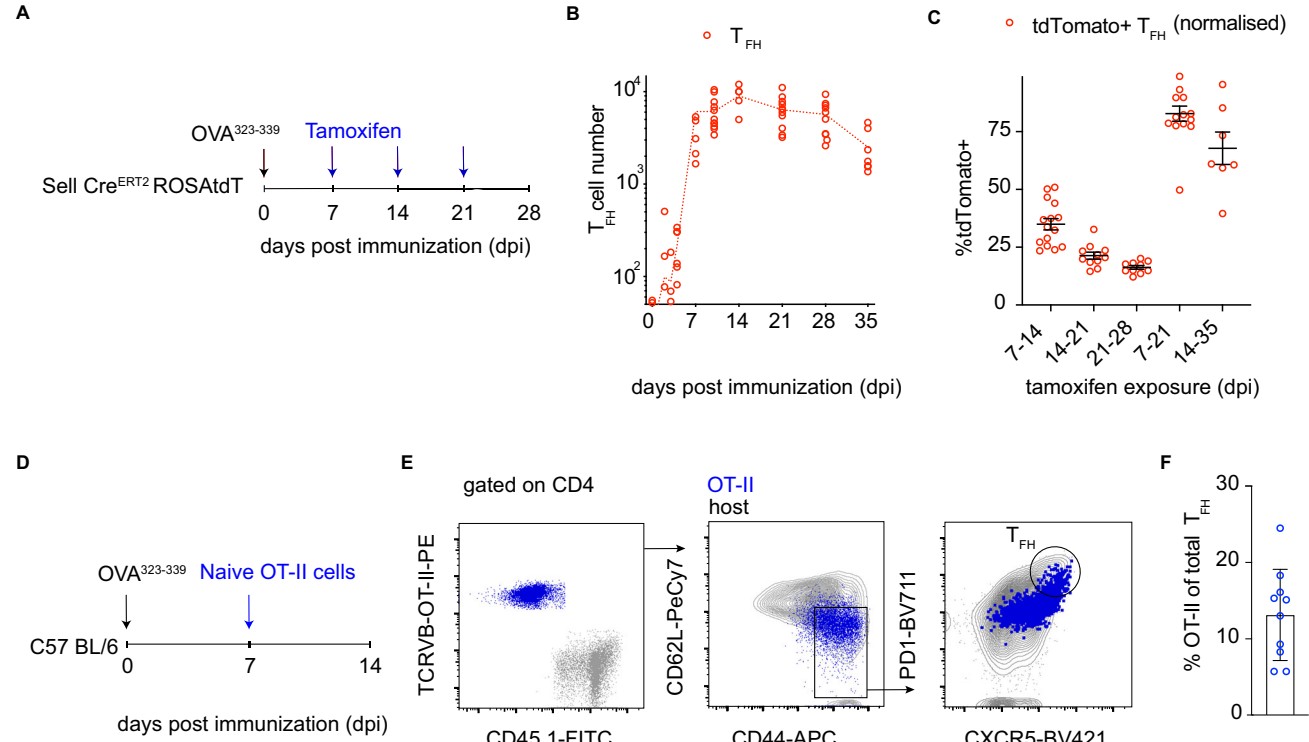

**Fig. 2 | Naïve polyclonal T cells can join ongoing immune responses.**
**A** Schematic representation of the experimental setup in (**B**, **C**). **B** Graph shows the kinetics of $T_{FH}$ cell development after immunization with OVA$^{323-339}$ peptide in adjuvant. The *y*-axis depicts the absolute numbers of $T_{FH}$ cells in individual popliteal lymph nodes (red) and the *x*-axis days after immunization. *n* = 3–15 per time point; each dot represents one lymph node. Data are presented as mean values. **C** Plots show the contribution of new invading cells over time. The *y*-axis is the frequency of tdTomato+ cells in $T_{FH}$ cells normalized to labeling in the naïve compartment. The *x*-axis is the interval between tamoxifen exposure and assay in days post immunization. *n* = 7–15 per time point; each dot represents one lymph node. Data are presented as mean value ± SEM. **D** Schematic representation of the experimental setup in (**E**, **F**). **E** Representative flow cytometry plots profiling phenotype of host or transferred OT-II T cells on day 14 post immunization. **F** Plots show the contribution of OT-II cells to total $T_{FH}$, on day 14 post immunization with OVA$^{323-339}$ when they were transferred 7 days prior. *n* = 10, each dot represents one mouse. Data are presented as mean value ± SD.

develop into $T_{FH}$ cells after day 7, but by day 21, they would be distinguished from resident $T_{FH}$ cells by tdTomato expression. Importantly, as GCs do not spontaneously form in popliteal LNs, unlike in the spleen, all $T_{FH}$ cells assayed from LNs are immunization-induced (Supplementary Fig. 3C). Notably, 21 days after immunization, 55% (72% when normalized to the maximal labeling achieved in naïve T cells from the same mouse) of $T_{FH}$ cells expressed tdTomato suggesting that a sizeable fraction of $T_{FH}$ cells develop from naïve T cells after the first 7 days of the reaction (Fig. 1E and Supplementary Fig. 3B)

We considered that the recruitment of new $T_{FH}$ cells into ongoing responses might be due to novel epitopes emerging from a complex protein antigen over time. Immunization with a single peptide in adjuvant limits epitope diversity and narrows the clonotypic response (Supplementary Fig. 3C, D). Therefore, we immunized CD62L reporter mice with a single OVA-derived 15-mer peptide: OVA$^{323-339}$ (Fig. 2A). GCs and $T_{FH}$ were detected by day 5 after immunization and peaked between day 10-14 before contracting thereafter (Fig. 2B). To examine the kinetics of naïve T cell differentiation into $T_{FH}$ throughout the response, tamoxifen was administered to CD62L reporter mice on days 7, 14, or 21 after immunization and assays performed 7 days later (Fig. 2A, C). The overall normalized mean fraction of tdTomato labeled $T_{FH}$ cells was 35%, 21%, and 16% on days 14, 21, and 28, respectively, showing continuous recruitment by naïve and recently activated T cells occurs throughout the response albeit with ingress decreasing over time (Fig. 2C).

To examine the overall contribution of newly arriving T cells to the $T_{FH}$ compartment after OVA$^{323-339}$ immunization, we measured the fraction of $T_{FH}$ cells labeled with tdTomato 14 days after tamoxifen administration (Fig. 2C). When tamoxifen was administered on days 7

or 14 after immunization, 82% and 67% (normalized) of all $T_{FH}$ cells were of new invader origin on days 21 and 35 respectively (Fig. 2C). We conclude that, despite decreased GC invasion overtime, newly developing $T_{FH}$ cells make a major contribution to the overall $T_{FH}$ cell compartment throughout the GC reaction.

CD62L can be expressed in T central memory ($T_{CM}$) in addition to naïve T cells (Supplementary Fig. 3E, F). To be sure that naïve conventional T cells can be recruited throughout the responses, preimmunized mice received naïve OT-II T cells on day 7 post immunization (Fig. 2D). Cells were followed to understand if they could be subsequently primed and recruited in the $T_{FH}$ compartment (Fig. 2D–F). On day 14 post immunization and 7 days after transfer, naïve OT-II had successfully differentiated into $T_{FH}$ and represented ~15% of the total $T_{FH}$ population (Fig. 2F). These data indicate that naïve T cells can be asynchronously primed and recruited throughout the response and so could provide a significant source of invading cells[31].

$T_{FH}$ cells exist both outside and inside of GCs[29,32]. To determine whether the $T_{FH}$ that develop throughout immune responses can enter ongoing GCs, we performed adoptive transfer experiments in TCR-beta deficient mice which lack endogenous T cells and directly visualized GCs by two-photon microscopy (Fig. 3A). CD62L reporter T cells and mCyan-expressing B18hi B cells that specifically recognize NP were adoptively transferred into TCR-beta deficient mice that were subsequently immunized with NP-OVA (Fig. 3A and Supplementary Fig. 4A–E). Recipient mice were treated with a lowered dose of tamoxifen 10 days after immunization to permit single T cell visualization by microscopy (Fig. 3B, C and Supplementary Fig. 4B–D). Immunization-induced GCs in popliteal lymph nodes imaged 7 days after tamoxifen injection showed tdTomato expressing invader $T_{FH}$

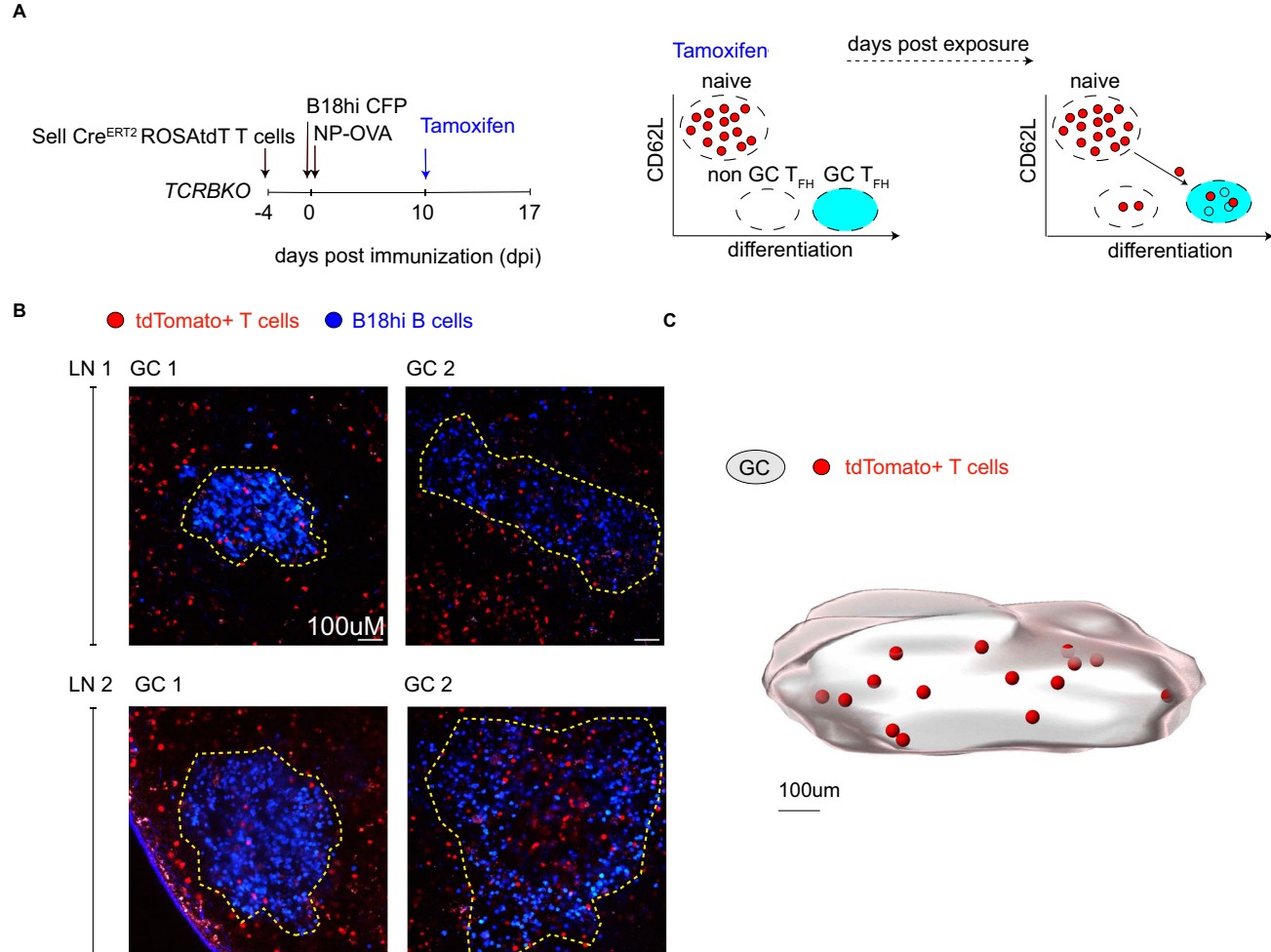

**Fig. 3 | Newly differentiated T$_{FH}$ cells join ongoing GC reactions. A** Schematic (left) and diagrammatic (right) representations of the experimental setup and labeling strategies used to visualize naïve T cell invasion of GC reactions. **B** Multiphoton Z-stack images to show individual GCs marked with yellow dashed lines. B18hi B cells (blue) and fate-mapped T cells (red). Data are from 2 lymph nodes from two biologically independent samples, and a total of 5 individual GCs were imaged ($n$ = 5). **C** Computational rendering (Imaris cell imaging software) of one of the GCs imaged from lymph node 1. The GC volume is defined by B18hi B cells. Invading T cells are marked as red spheres (renderings).

cells within individual GCs (Fig. 3B, C, Supplementary Fig. 4E). Thus, naïve T cells are continually recruited into the GC reaction, where they may participate in B cell selection.

To determine how newly invading T$_{FH}$ cells might differ from founder T$_{FH}$ cells, we purified both populations from the popliteal lymph nodes of OVA[323–339] immunized mice and performed single-cell mRNA sequencing (scRNAseq) (Fig. 4A–C and Supplementary Fig. 5A–C). Tamoxifen was administered to CD62L reporter mice on day 10 after immunization, and the 2 populations of T$_{FH}$ cells were purified 7 days later based on tdTomato expression (Supplementary Fig. 5A, B). TCR alpha and -beta chain sequences revealed tdTomato positive and negative T$_{FH}$ cell populations were significantly different from each other but similarly diverse and clonally expanded ($p < 0.001$ Supplementary Fig. 5C). Thus, naïve T cells developing into T$_{FH}$ cells undergo the same degree of clonal expansion as GC founder T cells and also remain clonally diverse.

TCR analysis revealed clones that were unique to the tdTomato positive and negative compartments or shared between the two (Fig. 4B, C). We reasoned that clonal overlap could be due to incomplete fate-mapping in CD62L reporter mice leading to tdTomato mosaicism in some clonal families. To mitigate this potential confounder, we excluded shared clones and singlets from our transcriptional analysis. Three distinct subpopulations of follicular T cells

(T$_F$), clusters 0, 1, and 2 were then identified using the single-cell gene expression profiles by Uniform Manifold Approximation and Projection (UMAP) (Fig. 4D). Cells in cluster 0 were enriched in genes associated with TCR signaling, including *Themis, Tcf7 and Ms4a4b* and appear to represent conventional T$_{FH}$ cells (Fig. 4D, E, G). Cluster 1 was enriched in genes associated with T$_{FR}$ function, including *Foxp3, Helios, CTLA4*, and *IL1R2*, and cluster 2 with genes associated with cellular stress and exhaustion, including *Hif1a, Cxxc5, Gadd45b, Pdcd1*, and *Fos* (Fig. 4D, G). Founder and new invader T$_F$ cells contained similar proportions of cluster 0 cells, indicating that both founder and new invader cells were actively participating in the GC reaction (Fig. 4E–G). Notably, cluster 2, which contains exhausted cells, was almost exclusively contributed to by founder T$_F$, which would be consistent with their prolonged residence in the GC. In contrast, the majority of invader T$_F$ cells found 17 days after immunization belonged to cluster 1, which has a signature associated with T$_{FR}$ cells (Fig. 4D–G). A comparison of individual gene expression profiles between invaders and founder cells revealed that the master transcription factor of regulatory cells, *Foxp3*, was among the most upregulated genes in the new invader compartment (Fig. 4H, red). Notably, *Foxp3* was similarly differentially regulated between tdTomato labeled or unlabeled clones when shared clones were included in the analysis.

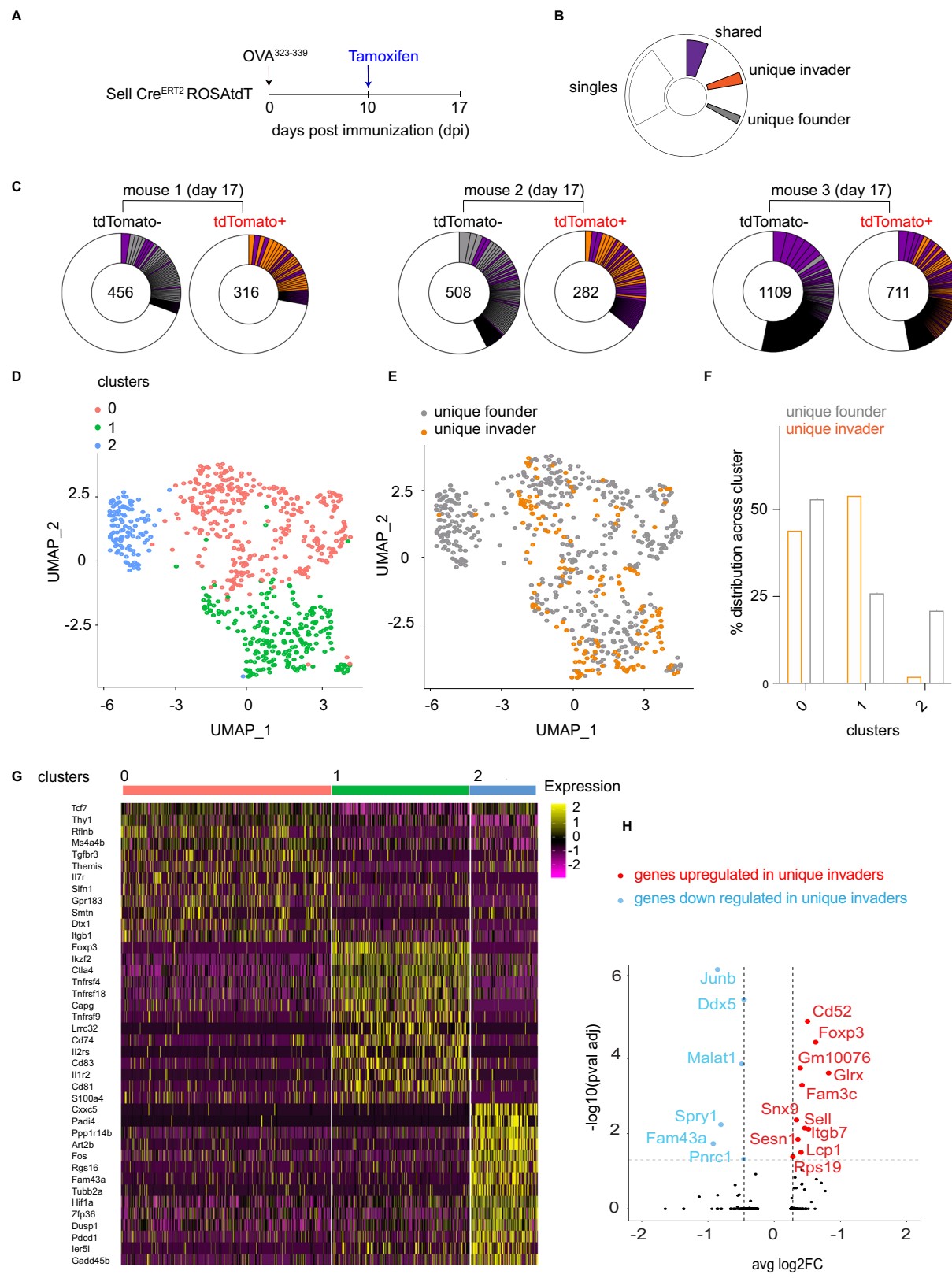

To confirm our transcriptional findings and to determine whether the newly invading $T_F$ cells had a higher proportion of cells expressing Foxp3 protein, we performed flow cytometry experiments using anti-Foxp3 antibodies (Supplementary Fig. 6A–C). To this end, CD62L reporter mice were immunized with OVA$^{323-339}$ peptides, tamoxifen was administered on day 10 after immunization, and flow cytometry was performed 7 days later (Supplementary Fig. 6A, B). In accordance with the differences identified by scRNAseq, newly recruited $T_F$ were enriched in Foxp3 protein as compared to founder cells (Supplementary Fig. 6B–D).

The naive T cell compartment contains thymic regulatory (tTregs) Foxp3$^+$ T cells as well as conventional Foxp3$^-$ T cells. Administration of

**Fig. 4 | Naïve cells that invade late GCs have a regulatory phenotype.**
**A** Schematic representation of the experimental setup used throughout. **B** Color-coded indexing for the distribution of shared (purple), unique invader (orange) or founder (gray) clones and singles (white). **C** Pie charts show the clonality among tdTomato positive and tdTomato negative $T_{FH}$ on day 17 post immunization. Segments report the proportional representation of each clone. Numbers inside the pie charts indicate the number of TCR sequences illustrated. **D** UMAP obtained from unique founder and unique invader $T_{FH}$ cells. **E** UMAP shows unique founders (gray) and unique invaders (orange). **F** Bar graph shows the relative distribution of unique founder (gray) or invader (orange) populations in the three clusters. *p*-value = 2.787e−15 by Fisher's Exact Test (two-sided). **G** Heat-map of genes that are differentially expressed between clusters 0,1,2. **H** Volcano plot shows the statistical significance (*p*-value) versus magnitude of change (fold change) of genes up (red) or downregulated (blue) in invaders. Positive values indicate that the gene is more highly expressed in invaders. These data are from 4 mice (*n* = 4) and 8 individual lymph nodes (sample size = 7), and technical replicates were performed. Adjusted *p*-value, based on Bonferroni correction using all genes in the dataset. The data discussed have been deposited in the NCBI Gene Expression Omnibus and are accessible through GEO series accession number: GSE240730.

tamoxifen to CD62L reporter mice labels both of these naïve T cell types. To determine whether late invading T cells showing a $T_{FR}$ phenotype are derived from conventional and/or regulatory naïve T cell compartments, we examined the Foxp3 expression by invaders that had entered the reaction between days 10–17 of immunization (new Supplementary Fig. 7A–C). Among the invaders, 38 clonal families contained at least one member that expressed Foxp3 transcripts. Of these 38 expanded clones, 25 were uniformly Foxp3⁺ suggesting that they were derived from a precursor that expressed Foxp3 before entering the $T_F$ compartment (Supplementary Fig. 7A-C). The remaining 13 clonal families contained both Foxp3⁺ and Foxp3⁻ cells suggesting that some $T_{FR}$ cells were derived from a Foxp3⁻ precursor that gave rise to daughters, some of which expressed Foxp3. We conclude that both conventional and tTreg cells can differentiate from naïve to $T_{FR}$ cells throughout the GC reaction.

$T_{FR}$ cells inhibit plasma cell development, IgE and IgA production and protect against the emergence of autoantibodies; in addition, they appear to help mediate GC termination[23,24,33]. Consistent with these observations, the frequency of Foxp3⁺ positive cells among $T_F$ cells remained low in the first 14 days after immunization with OVA[323–339] but increased thereafter in parallel with GC B cell and $T_F$ cell contraction (Fig. 5A, B and Fig. 2B). To better understand the role of $T_{FR}$ accumulation, we utilized mice in which $T_{FR}$ cells can be temporally depleted[22]. Briefly, these mice express a $Cxcr5^{IRES-LoxP-STOP-LoxP-DTR}$ allele combined with a $FoxP3^{IRES-CreYFP}$ allele which drives the expression of diphtheria toxin receptor (DTR) in $T_{FR}$ cells allowing for their depletion upon diphtheria toxin (DT) administration ($T_{FR}$-DTR mice). $T_{FR}$-DTR mice and C57 BL/6 wildtype (wt) mice were immunized with OVA[323–339], and DT was subsequently administered on days 10 and 13, resulting in $T_{FR}$ depletion in conditional mice. GC-associated responses were then assayed on day 15 post immunization (Fig. 5C–E). In $T_{FR}$-depleted mice, there was a trend towards an increase in total $T_F$ cell numbers as compared to immunized conditional mice that didn't receive DT or immunized and DT treated wt controls. (Fig. 5E)[22]. $T_{FR}$-depleted mice also had significantly increased GC responses, as ascertained by total GC B cell numbers compared to both control groups (Fig. 5E). This suggested $T_{FR}$ plays a role in controlling the magnitude and duration of the GC response. Together these results indicate that increases in $T_{FR}$ frequencies over time aid GC involution, while delayed accumulation of $T_{FR}$ extends GC reactions.

To examine how late invader T cells contribute to the overall accumulation of $T_{FR}$ throughout the GC reaction, we immunized CD62L reporter with OVA[323–339] peptide, administered tamoxifen on days −3, 7, 10, and 20 and performed flow cytometry 7 days later (Fig. 5F–H). During the first 14 days of the GC reaction, resident and newly recruited $T_F$ contained similar proportions of Foxp3⁺ cells (Fig. 5G). However, on days 17 and 27, when the GC undergoes involution, the proportion of Foxp3⁺ cells among newly developing $T_F$ cells was greater than among the founders (*p* < 0.001 and *p* = 0.01 Fig. 5G). To examine the proportional contribution of newly invading T cells to the $T_F$ Foxp3+ compartment, tamoxifen was administered on day 10 after immunization, and Foxp3 protein expression was assayed 14 days later (Fig. 5H). Fate-labeled cells comprised 73% (normalized) of the Foxp3⁺ $T_F$ compartment suggesting $T_{FR}$ are largely comprised of new invaders in the late stages of the GC reaction (Fig. 5H). This data suggests that the accumulation of $T_{FR}$ cells during late-stage GCs is heavily contributed to by the invading compartment.

Slow antigen delivery immunization regimes have been shown to lead to longer-lasting GCs with more $T_{FH}$, but the underlying mechanism remains unknown[34]. To determine how antigen availability might affect $T_{FR}$ accumulation, we compared a single bolus immunization with slow delivery antigen dosing over 5 days[34,35] (Fig. 6A). Bolus immunization results in rapid peak antigen concentrations followed by equally rapid antigen clearance[34–36]. In contrast, slow antigen delivery results in prolonged antigen retention in the form of immune complexes that develop as a result of specific antibody production over time[36]. CD62L reporter mice were immunized with either 1 dose of 20 micrograms of OVA[323–339] peptide or the same total amount of peptide dosed in escalating amounts over 5 days (Fig. 6A)[37]. Tamoxifen was administered on day 10 after immunization, and GCs were examined 7 days later (Fig. 6A–D). Compared to bolus immunization, escalating doses of antigen resulted in increased numbers of GC B cells and $T_{FH}$ cells, albeit lower numbers of $T_{FR}$ cells (Fig. 6B)[37]. Notably, the proportion of Foxp3⁺ cells among total $T_F$ was only 9% in reporter mice receiving escalating dose immunization compared to 44% in bolus immunized mice, and this difference was due in large part to invaders (Fig. 6C, D). Thus, the composition of invading cells appears to be regulated dynamically by antigen availability. Consistent with this idea, in GCs from the mesenteric lymph nodes and Peyer's patches, which are chronically exposed to gut antigens, the frequency of Foxp3⁺ cells in invading compartments never exceeds 15% (Supplementary Fig. 8A, B).

Immunization with mRNA-based vaccines also results in robust and prolonged GCs (Supplementary Fig. 8C)[38,39]. To understand if delayed $T_{FR}$ accumulation is a feature of mRNA-based vaccines, CD62L reporter mice were immunized with the COVID-19 BioNTech (Pfizer), tamoxifen was administered on days 10, 14 and 24 and 35, and flow cytometry was performed 7 days later (Fig. 6E). At each time point analyzed, tdTomato+ (invader) cells continued to contribute to the total $T_F$ compartment, confirming that ongoing recruitment occurs in this immunization setting (Fig. 6F). Despite this, the frequency of $T_{FR}$ cells among tdTomato+ cells at day 17 post immunization were significantly lower in mice immunized with BioNTech (Pfizer) as compared to OVA[323–339] (Fig. 6G). However, following day 21 of BioNTech (Pfizer) immunization, when the GC response is waning, the fraction of inhibitory $T_{FR}$ increases dramatically relative to $T_{FH}$ over time in the invading compartment (Fig. 6H and Supplementary Fig. 7C). This data suggests that delayed $T_{FR}$ recruitment into the GC response is a feature of BioNTech (Pfizer) immunization and might play a role in supporting longer GC reactions.

## Discussion

B cells diversify their antibody genes in the GC responses by somatic hypermutation. Although TCRs are not subject to hypermutation, the GC T cell compartment diversifies by selective clonal expansion and contraction over time[27]. Our experiments indicate that naïve conventional and tTreg T cell immigration into the GC reaction makes a major

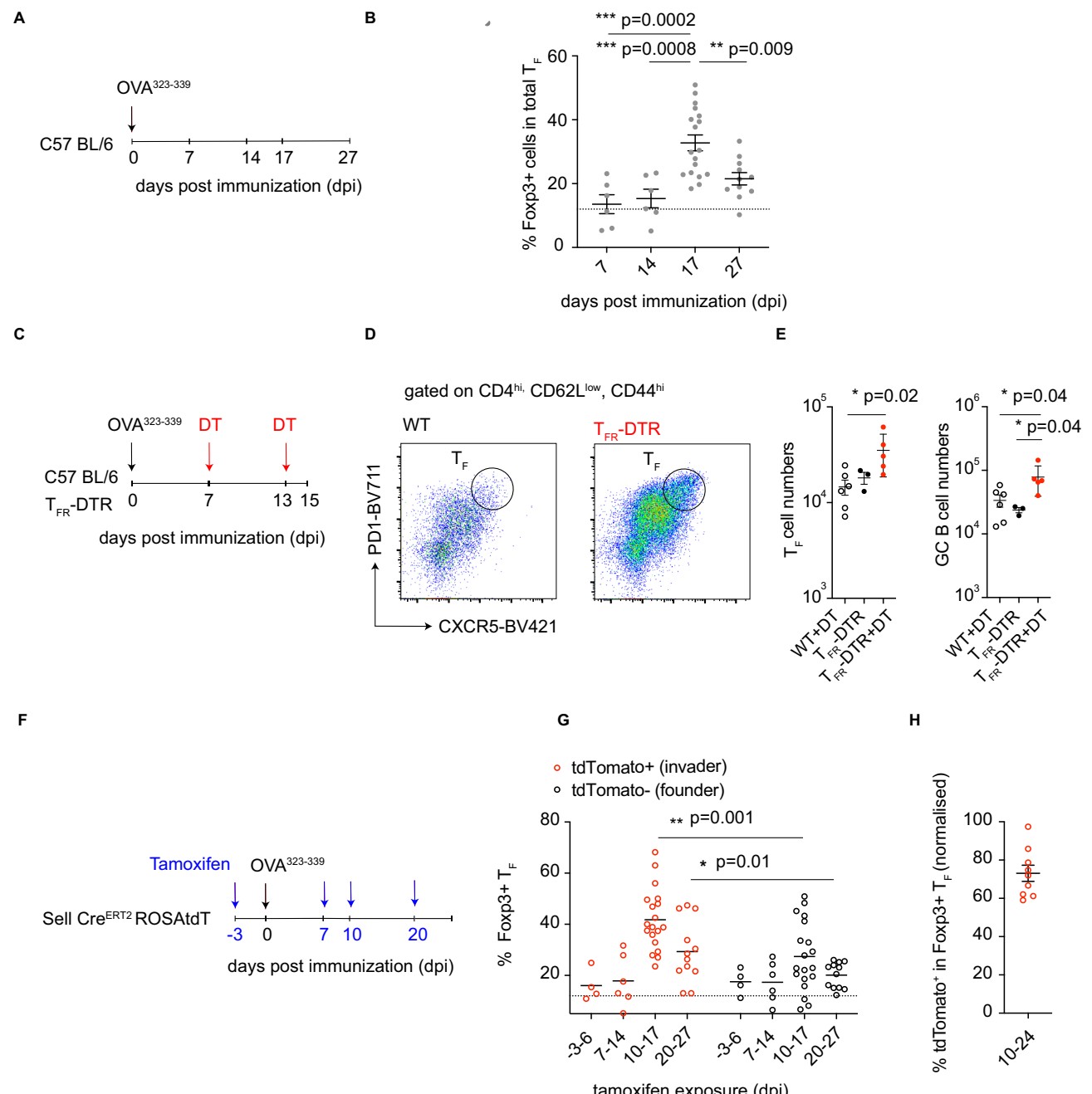

**Fig. 5 | T_FR accumulates in late-stage GC reactions. A** Schematic representation of the experimental setup used in (**B**). **B** Kinetics of Foxp3 expression among T_F cells following immunization with OVA^323–339 peptide in adjuvant. The *y*-axis depicts the frequency of Foxp3+ cells among T_F cells, and the *x*-axis is days after immunization. The dotted line shows the average frequency of Foxp3+ cells among naive CD4+ T cells. *n* = 6–18 per time point, each dot represents one lymph node from an individual mouse. Data are presented as mean value ± SEM *** *p*-value = 0.0002, *** *p*-value = 0.0008, ****p*-value = 0.009, calculated by ANOVA with multiple comparisons. **C** Schematic representation of the experimental setup used in (**D, E**). **D** Representative flow cytometry plots profiling the frequency of T_FH cells in mice from the respective experimental groups (right). (left) Plots show the frequency of T_FH (*y*-axis) cells among CD4^hi, CD62^low, and CD44^hi cells between experimental groups (*x*-axis) on day 15 post immunization. **E** Plots show the total number of T_F cells (left) or the total number of GC B cells (right) calculated (*y*-axis) between the experimental groups (*x*-axis). *n* = 3–6 mice per group; each dot represents one mouse. Data are presented as mean value ± SD. *p*-value = 0.02 and *p*-value = 0.04 calculated by ANOVA with multiple comparisons. **F** Schematic representation of the experimental setup used in (**G, H**). **G** Plots depict the kinetics of Foxp3 expression among invader or founder T_F cells. The *y*-axis depicts the frequency of Foxp3+ cells among invader (red) or founder (black) T_F cells at the time points after immunization (*x*-axis). *n* = 4–19 mice per timepoint, each dot represents a single lymph node. The data presented are the mean. ***p*-values = 0.001 and *p*-value = 0.01 calculated by unpaired Student's *t*-test. **H** Plot shows the normalized relative proportion of tdTomato+ cells among T_FR cells. *n* = 9, and the data presented are the mean ± SEM.

contribution to the diversification and clonal remodeling of the T follicular compartment during an immune response.

Limiting the number of T_FH in the GC is essential to maintain stringent antibody affinity selection and to avert the emergence of autoantibody-producing B cells[19–22,25,26]. Nevertheless, some T_FH cells

undergo clonal expansion in response to antigen, others contract or even become extinct, while novel clones emerge overtime[27]. We find that most of the T cell clones in the late GC are novel. These new clones could originate from previously undetected founder T cells or from new invaders entering in the later stages of the reaction. Our

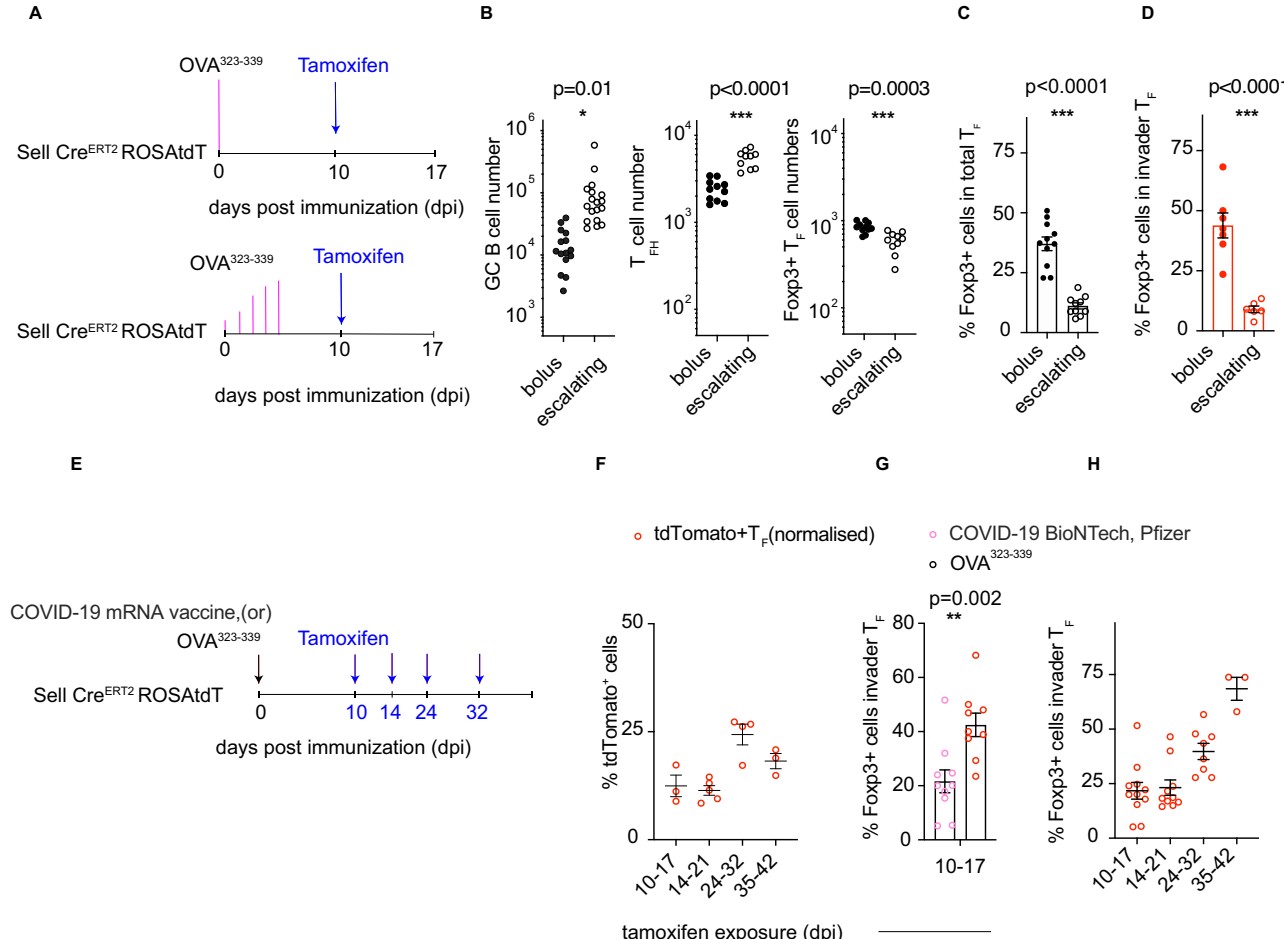

**Fig. 6 | Antigen-dependent T_FR development. A** Schematic shows the bolus (top) and slow delivery (bottom) immunization strategies, tamoxifen delivery, and sampling schedules used in (**B**–**D**). **B** Bar graphs show the total number of GC B cells (left), $T_{FH}$ cells (center), and Foxp3+ $T_{FR}$ cells (right) in popliteal lymph nodes after bolus or escalating dose immunization. *$p$-value = 0.01, ***$p$-value < 0.0001, $p$-value = 0.0003 by unpaired Student's $t$ test. $n$ = 9–18, and each dot represents a single lymph node. **C** Bar graphs show the percentage of Foxp3+ cells among total $T_F$ cells after bolus and escalating dose immunization. $p$-value < 0.0001 by unpaired Student's $t$-test (two-tailed). $n$ = 10–11 per group, and each dot represents a single lymph node. **D** Bar graphs show the percentage of Foxp3+ cell invaders after bolus and escalating dose immunization. $p$-value < 0.0001 by unpaired Student's $t$-test (two-tailed). $n$ = 6–7 per group, and each dot represents a single lymph node. **E** The schematic shows the experimental setup used for (**F**–**H**). **F** Plot shows the normalized relative proportion of tdTomato+ cells among $T_F$ cells. Each dot is an individual draining inguinal lymph node from the site of immunization with the COVID-19 BioNTech (Pfizer) vaccine. $n$ = 3–4 per timepoint, and the data presented are the mean ± SD. **G** Bar graphs show the frequency of Foxp3+ cells among total $T_F$ cells in invaders cells on day 17 post immunization. The $x$-axis is the interval between tamoxifen exposure and assay in days post immunization. **$p$-value = 0.002 by unpaired Student's $t$-test (two-tailed). $n$ = 10–11 per group, and the data presented are the mean ± SEM. **H** Plots depict the kinetics of Foxp3 expression among invader $T_F$ cells from inguinal lymph nodes from COVID-19 BioNTech (Pfizer) vaccinated mice. The $y$-axis depicts the frequency of Foxp3+ cells among invader $T_F$ cells at the time points after immunization ($x$-axis). $n$ = 3–11 per timepoint, and the data presented are mean ± SEM.

experiments indicate that naïve invaders that are asynchronously primed and recruited throughout the response, make up the majority of the follicular T cells in the late stages of the immune response. This influx of novel $T_F$ clones parallels an influx of naïve B cells that develop into GC cells throughout the response[28]. Although the source of antigen that feeds continual B cell activation is not known, p-MHCII expressed on the surface of activated B cells would be a natural source of cognate antigen for naïve T cell activation throughout the response. Indeed, activated B and T cells form cognate antigen-dependent pairs that subsequently enter the GC reaction together, and GC B cell antigen presentation is essential for $T_{FH}$ responses[6,11,28,33,40–43].

In addition to dynamic clonal replacement over time, the $T_{FH}$ compartment also undergoes phenotypic and functional changes. Notably, the relative fraction of inhibitory $T_{FR}$ increases relative to $T_{FH}$ longitudinally reaching a peak during GC involution[23,24,33]. Loss of $T_{FR}$ cells is associated with increased autoantibody production, altered B cell selection, increased class switching to IgE and IgA, and increased

plasma cell production[19–22]. Thus, the dynamic equilibrium between $T_{FH}$ and $T_{FR}$ is essential to shaping physiologic humoral immune responses.

While late surges of regulatory $T_{FR}$ cells have been documented, the precise mechanism responsible for these changes is not known. Our experiments show late invading T cells are major contributors to $T_{FR}$ accumulation because they make up the majority of the $T_{FR}$ compartment in the late GC. Hence invasion plays a pivotal role in modulating the $T_{FH}$:$T_{FR}$ ratio in ongoing GC reactions. Additional mechanisms, including phenotypic switching of $T_{FH}$ into Foxp3 expressing cells, clonal expansion of pre-existing $T_{FR}$, and differential cell death in the late GC, may also contribute to alterations in the ratio of $T_{FH}$ and $T_{FR}$[23,41].

GC contraction is associated with decreasing antigen availability. Prolonging antigen availability by escalating slow delivery immunization extends the GC reaction and increases the number of participating GC T and B cells[34,37]. Crucially, these long-lasting GCs

allow for sustained somatic hypermutation and so hold much promise for the development of antibodies to difficult vaccine targets such as HIV-1 that require high levels of somatic mutation. We find that delivering antigen through escalating low-dose immunization, as opposed to bolus, acts to curb $T_{FR}$s and favors $T_{FH}$ cell development by new resulting in larger and prolonged GC B cell responses. Conversely, the more rapid decrease in antigen concentration in bolus-immunized mice naturally favors $T_{FR}$ development, limiting constitutive T help and consequently favoring GC contraction. Similarly, mRNA-based vaccines induce extended and durable GC reactions. The role of the corresponding $T_F$ repertoire in this phenomenon has not yet been fully explored, but our experiments suggest that delayed $T_{FR}$ recruitment might contribute to prolonged GCs that support additional rounds of affinity selection and plasma cell development. In conclusion, the fate mapping experiments reveal a previously unappreciated contribution of naïve T cells to the diversification of the T follicular cell compartment throughout the reaction. Thus continual invasion of the GC alters the balance of $T_{FH}$ and $T_{FR}$ in an antigen-dependent manner and thereby contributes to GC size, selection, and longevity.

## Methods

### Mice
Mice were housed at a temperature of 72 °F and humidity of 30–70% in a 12-h light/dark cycle with *ad libitum* access to food and water. Male and female mice aged 8–10 weeks at the start of the experiment were used throughout. C57BL/6 J (Jackson strain #:000664), B6.129P2-*Tcrb*[tm1Mom]/J mice (Jackson strain #:002118), OT-II (C57BL/6 J) mice were purchased from Jackson Laboratories. *OTII*, OTII PAGFP, and PAGFP mice were generated and maintained at Rockefeller University. *Sell*-CreERT2 ROSAtdT reporter mice were generated in B6 ES cells and exclusively crossed to B6 animals for 10 generations and maintained at Rockefeller University. $T_{FR}$-DTR mice were gifts from the Peter Sage laboratory, where they were generated and characterized[22]. All mouse experiments were performed under Institutional Review Board-approved protocols. Sample sizes were not calculated a priori. Given the nature of the comparisons, mice were not randomized into each experimental group, and investigators were not blinded to group allocation.

### Immunizations and treatments
C57BL/6 J, *Sell*CreERT2 ROSAtdT, and B6.129P2-*Tcrb*[tm1Mom]/J mice (6–12 weeks old) were immunized with 20 μg or 50 μg of NP17–OVA (Biosearch Technologies) precipitated in alum in footpads or intraperitoneally respectively. For OVA[323–339] peptide immunizations, recipient mice received a footpad injection of a total of 20 μg of synthetic peptide mixed in Sigma Adjuvant System® (adjuvant was used to provide the necessary stimulus to the immune system to allow CD4 T cell responses). For escalating dose experiments, mice were immunized over 5 days with 2–6 μg of peptide. Adjuvant only accompanied peptide on the first injection. In some cases, mice received 1 μg of diphtheria toxin in PBS i.p. to delete $T_{FR}$ cells at indicated time points. For COVID-19 BioNTech (Pfizer) immunization experiments, C57BL/6 J and *Sell*CreERT2 ROSAtdT mice received 1ug of vaccine intramuscularly and the associated inguinal lymph nodes were harvested at respective time points.

Activation of the Cre recombinase in the CD62L reporter mice was induced by one oral administration of 12 mg or 6 mg (in two-photon imaging experiments) of tamoxifen (T5648; Sigma) in 200 μl of corn oil (C8267; Sigma) at the indicated time points.

### Hemi-splenectomy
Mice were kept on antibiotics as prophylaxis against infection following surgical intervention. On d7 post immunization, mice were anesthetized with isoflurane. The left side of the mouse was shaved and cleaned before an incision was made in the skin, followed by a smaller incision in the peritoneal wall to allow access to the spleen. The section of spleen to be removed was tied off by using sutures to prevent bleeding and then cut out while leaving the splenic artery intact. The peritoneal wall was closed and stitched using perma-hand silk 5-0 sutures (Ethicon). The skin was closed using 9 mm wound clips (Clay Adams brand, Becton Dickinson). Following recovery from anesthesia, mice were transferred to a new clean cage with a heating pad.

### Peptide Synthesis
The OVA[323–339] peptide was synthesized and then purchased from Anaspec or Genscript.

### T cell transfer
Single-cell suspensions were prepared from the spleens and lymph nodes of donor mice.

CD4+ and CD8+ T cells were enriched from spleens by magnetic bead selection (StemCell Technologies). Total T cells from 1 mouse equivalent ($30 \times 10^6$) and $5 \times 10^6$ B18hi B cells were injected into recipient B6.129P2-*Tcrb*[tm1Mom]/J mice by intravenous injection. In this setting, only the missing lymphocyte population (T cells) was properly reconstituted, creating T-cell-specific reporter mice.

### Flow cytometry
Single-cell suspensions were stained with antibodies directly conjugated to surface markers. Intracellular stains were performed using commercially available Fix and permeabilization solutions coupled to incubation with Foxp3 antibodies. Multi-color cytometry was performed on the Symphony flow cytometer (BD Biosciences) and analyzed with FlowJo v10.4.2.

### Microscopy
LNs were harvested and cleared of adipose tissue under a dissecting microscope and placed in PBS between two coverslips held together by vacuum grease. Throughout the preparation and imaging, the tissue was kept on a cooled metal block. Multiphoton imaging was performed as described (Mesin, 2020) using an Olympus FV1000 upright microscope fitted with a 25 × 1.05NA Plan water-immersion objective and a Mai-Tai DeepSee Ti-Sapphire laser (Spectraphysics). In adoptive transfer experiments, $50 \times 10^6$ T cells and $5 \times 10^6$ B18 donor B cells were transferred.

### RNA sequencing
For single-cell RNA sequencing, single-cell suspensions were prepared from half-spleens of NP-OVA immunized *Sell*CreERT2 ROSAtdT mice on days 7 and 21 after immunization. Samples were indexed with TotalSeqC (BioLegend) cell surface antibodies, and CD4+, CD62L[low], CD44[hi], PD1[hi], CXCR5[high], tdTomato+ Tfh cells were purified by flow cytometry, pooled, and loaded onto a Chromium Controller (10x Genomics). Single-cell RNA-seq libraries were prepared using the Chromium Single Cell 5′ v2 Reagent Kit (10× Genomics) according to the manufacturer's protocol. Libraries were loaded onto an Illumina NextSeq with the mid-Output Kit (150 paired end) for V-D-J analysis or NOVAseq for single-cell gene expression. Hashtag indexing was used to demultiplex the sequencing data and generate gene-barcode matrices, respectively.

### Statistical analyses
Statistical tests were conducted using Prism (GraphPad) software. Unpaired, two-tailed Student's *t*-tests and one-way ANOVA with Tukey's post hoc tests to further examine pairwise differences were used. Data were considered statistically significant at $*p \leq 0.05$, $**p \leq 0.01$, $***p \leq 0.001$, and $****p \leq 0.0001$. The number of mice per group, the number of replicates, and the nature of error bars are

indicated in the legend of each figure. Center bars always indicate the mean.

## Data analysis

We used Cell Ranger (v.6.0.1) from 10× Genomics for single-cell UMI quantification and TCR clonotype assembly. Hashtag-oligos (HTOs) UMI counts were processed using CITE-Seq-Count (v.1.4.3). We used Seurat (v.4.0.0), an R package to analyze single-cell RNA-seq data, to identify differentially expressed genes. We defined the invaders and residents Tfh clones by detecting cells exclusively present in the tdTomato+ and tdTomato- compartments, respectively. Genes expressed in at least 10% of all cells belonging to invaders or resident clones, with the adjusted P value by Bonferroni correction less than 0.05. Pseudotime trajectory was performed by Monocle3 Seurat-Wrapper (v1.0.0) with cluster 1 set as the root. MacVector was used for sequence analysis. Graph Prism 9 was used for data analysis and for graph generation. Data collection Flow cytometry data was collected using FACDIVA version 8.0.2.

## Ethical statement

All procedures in mice were performed in accordance with protocols approved by the Rockefeller University IACUC. All animal experiments were performed according to the protocols approved by the Institutional Animal Care and Use Committee of NIAID, NIH. Carbon dioxide ($CO_2$) inhalation was used as the method of euthanasia

## Statistics and reproducibility

No statistical analysis was performed to predetermine sample size, but these are standard in the field. The Investigators were not blinded to allocation during experiments and outcome assessment. Each experiment was repeated >2 times with similar results, and the sample size (*n*) and statistical comparisons were annotated in the Figure legends throughout.

## Reporting summary

Further information on research design is available in the Nature Portfolio Reporting Summary linked to this article.

## Data availability

The authors declare that all data supporting the findings of this study are available within the article and its supplementary file or from the corresponding author upon reasonable request. The data discussed in this publication have been deposited in the NCBI Gene Expression Omnibus and are accessible through GEO series accession numbers: GSE147182 and GSE240730. Source data are provided in this paper.

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

## Acknowledgements

We thank Ervin E. Kara and Thomas Hagglof for the generation and maintenance of CD62L reporter mice, T. Eisenreich for help with mouse colony management, and technical help, B. Zhang and C. Zhao at The Rockefeller University Genomics Resource Center for assistance with 10× genomics and high-throughput sequencing, K. Gordon for assistance with cell sorting, and all members of the Nussenzweig laboratory for discussion. We thank Johanne T. Jacobsen for generating computational renderings using Imaris Cell imaging software. We thank Peter Sage for the generation and gift of the $T_{FR}$-DTR mice. This work was supported by NIH grant 5R37 AI037526 and NIH Center for HIV/AIDS Vaccine Immunology and Immunogen Discovery (CHAVID) 1UM1AI144462-01 to M.C.N. J.M. is a Branco Weiss fellow. M.C.N. is an HHMI investigator.

## Author contributions

J.M. and M.C.N. conceived, designed, and analyzed the experiments. J.M., R.M., M.U., and A.J.M. carried out all experiments. V.R. and T.Y.O. performed the bioinformatic analysis. C.R.N. contributed to microscopy experiments and discussions. J.M. and M.C.N. wrote the paper with input from all co-authors.

## Competing interests

The authors declare no competing interests.
