## [Peer Review File · Nature Communications]

Continually recruited naïve T cells contribute to the follicular helper and regulatory T cells pools in germinal centersEditorial Note: This manuscript has been previously reviewed at another journal that is not operating a transparent peer review scheme. This document only contains reviewer comments and rebuttal letters for versions considered at Nature Communications.

Reviewers' Comments:

Reviewer #1:

Remarks to the Author:

The authors have performed a number of experiments that have clarified and addressed the concerns raised. Their additional data have led them to conclude "that both conventional and tTreg cells can differentiate from naïve to TFR cells throughout the GC reaction". This needs to be reflected in the abstract, because in its current form, it is likely to mislead readers to believe naïve (Foxp3-) T cells rather than Tregs are exclusively giving rise to Tfr cells. It would be appropriate that the authors specifically acknowledge that Foxp3+ "naïve" Tregs together with naïve T cells are recruited and give rise to Tfr cells.

REVIEWERS' COMMENTS

- Reviewer #1 (Remarks to the Author):

The authors have performed a number of experiments that have clarified and addressed the concerns raised. Their additional data have led them to conclude "that both conventional and tTreg cells can differentiate from naïve to TFR cells throughout the GC reaction". This needs to be reflected in the abstract, because in its current form, it is likely to mislead readers to believe naïve (Foxp3-) T cells rather than Tregs are exclusively giving rise to Tfr cells. It would be appropriate that the authors specifically acknowledge that Foxp3+ "naïve" Tregs together with naïve T cells are recruited and give rise to Tfr cells.

- We thank the reviewer for their appreciation of the number of additional experiments we conducted to clarify the manuscript and address the concerns raised. We have taken the advice to further amend the abstract to reflect this. We have included the following sentence in the new abstract : "GC reactions continually recruit T cells from both the naïve conventional and naïve thymic regulatory T cell (Treg) repertoire".